# A Novel Wireless Low-Cost Inclinometer Made from Combining the Measurements of Multiple MEMS Gyroscopes and Accelerometers

**DOI:** 10.3390/s22155605

**Published:** 2022-07-27

**Authors:** Seyedmilad Komarizadehasl, Mahyad Komary, Ahmad Alahmad, José Antonio Lozano-Galant, Gonzalo Ramos, Jose Turmo

**Affiliations:** 1Department of Civil and Environment Engineering, Universitat Politècnica de Catalunya, BarcelonaTech. C/Jordi Girona 1-3, 08034 Barcelona, Spain; milad.komary@upc.edu (S.K.); mahyad.komarizadeh@upc.edu (M.K.); ahmad.alahmad@estudiantat.upc.edu (A.A.); gonzalo.ramos@upc.edu (G.R.); 2Department of Civil Engineering, Universidad de Castilla-La Mancha, Av. Camilo Jose Cela s/n, 13071 Ciudad Real, Spain; joseantonio.lozano@uclm.es

**Keywords:** low-cost sensors, NodeMCU, Allan variance, noise reduction, influence line measurement, structural health monitoring

## Abstract

Structural damage detection using inclinometers is getting wide attention from researchers. However, the high price of inclinometers limits this system to unique structures with a relatively high structural health monitoring (SHM) budget. This paper presents a novel low-cost inclinometer, the low-cost adaptable reliable angle-meter (LARA), which combines five gyroscopes and five accelerometers to measure inclination. LARA incorporates Internet of Things (IoT)-based microcontroller technology enabling wireless data streaming and free commercial software for data acquisition. This paper investigates the accuracy, resolution, Allan variance and standard deviation of LARA produced with a different number of combined circuits, including an accelerometer and a gyroscope. To validate the accuracy and resolution of the developed device, its results are compared with those obtained by numerical slope calculations and a commercial inclinometer (HI-INC) in laboratory conditions. The results of a load test experiment on a simple beam model show the high accuracy of LARA (0.003 degrees). The affordability and high accuracy of LARA make it applicable for structural damage detection on bridges using inclinometers.

## 1. Introduction

Structural health monitoring (SHM) has attracted the attention of engineers over past decades as a control system to measure the structural response of structural elements to prevent future potential failures in civil infrastructures. A number of factors and situations such as construction defects, fatigue and environmental factors might decrease the structure’s serviceability and safety over time [1,2,3]. Therefore, monitoring and assessing structures’ health state throughout their life cycle is essential to minimize the future reparation costs and to confirm the structural safety and serviceability [4,5]. SHM applications provide vital information about the actual structural response of infrastructures, the condition of the structures and their performance. As indicated by many scholars (such as [6,7]), SHM can be used to calibrate the simulated model of the real structures (digital-twin), which imitates the performance of the structures [8]. Digital twins can then be used to evaluate the decision-making alternatives during the maintenance phase of structures under study [9,10].

SHM measurements can be used to determine structural parameters with structural system identification techniques [11,12]. Structural system identification targets identifying the computer-based model’s parameters (such as axial or flexural stiffness) to estimate the structural response of the structure [13]. Based on the nature of the structural response and the features of the external excitation, structural system identification methods can be classified as static or dynamic [12,14].

In fact, environmental phenomena such as temperature and humidity could cause pathologies such as crack opening [15], rotations, settlements and corrosion in the structure [2,16,17]. The time variation of these changes is notably slow. Consequently, they are classified as static. On the other hand, those excitations that change instantly throughout time (such as traffic-induced vibrations, ambient activities and waves from seismic activities) lead to dynamic structural responses [18].

For measuring static and dynamic responses, sensors are widely used in SHM systems [19]. Accelerometers are commonly used for monitoring the dynamic response of the structures, while the most common sensors for static measurements include strain gauges, inclinometers and thermometers [20,21].

Accelerometers can estimate and identify a structure’s dynamic characteristics by measuring changes in the structural response [2,22,23]. Even though accelerometers can detect global structural damages to a structure, they traditionally fail to detect the damage location and its severity [24]. Displacement sensors such as laser displacement sensors (LDS) can be used in load tests to help to locate the damage and its extension as long as a particular reference point exists [25,26]. Unfortunately, a number of limitations on-site can make the proper definition of the required reference points difficult [27]. Alternative strain-type sensors can be used to evaluate the extent of the damage and its location. In fact, this type of sensor has shown remarkable accuracy and applicability in the literature [28,29,30]. However, a large number of this type of sensor might be needed to monitor the structure’s structural properties entirely [31].

In order to overcome the drawback of the aforementioned sensors, inclinometers can be used. Angular sensors (inclinometers, tilt sensors) are manufactured to estimate the angular rotation of a target object in respect to an artificial horizon [24]. Most inclinometers follow the principle of measuring responses induced by pendulum behavior due to gravity [32]. Furthermore, this slope can be used to calculate the drift of vertical members and vertical deflection of the horizontal elements [33].

In the past decades, inclinometer sensors have been widely used in a number of sectors. In the civil engineering industry, inclinometers were firstly introduced for geotechnical applications. Over the years, improvements in sensor accuracy have enabled its applications to other civil engineering fields such as bridge structural health monitoring [32,34]. Inclinometers have been widely implemented in the literature to study the structural response of bridges. For example, Glišić et al. [35] used long-gauge deformation sensors and inclinometers to analyze a post-tensioned concrete bridge during its construction, post-tensioning and first-year operation stages. The outcome was used in verifying the post-tensioning and health state of the bridge.

Literature review shows the important role of inclinometers [36] in the long-term monitoring of long-span (such as cable-stayed and suspension) bridges [37,38,39]. In addition, several scholars (such as [40,41,42]) reported the traditional use of inclinometers studying the boundary condition response of the bridge abutments. Another common use of these types of sensors is the calculation of bridge deck deflections [43,44,45].

Table 1 illustrates the characteristics of some of the commercially available inclinometers and is sorted by the price of the sensors. This information includes the measurement range, the resolution, the sampling rate and the cost of the introduced inclinometers. It should be noted that prices are based on recent producer declaration and are VAT excluded.

Analysis of Table 1 shows a wide range of prices (varying between €350 up to €3950) and measurement ranges (varying between 0.5 and 85.0 degrees). It can be seen that inclinometers with a lower range have a higher resolution and price. Furthermore, inclinometers with higher resolution typically have higher costs and lower sampling frequencies. In fact, some of the models above have been used in applications intended for measuring the structural responses of bridges. Examples of the applications of sensors listed in Table 1 include the use of Zerotronic inclinometer for validating modal calibration techniques with measured data of the Lutrive bridge in Switzerland [46].

Contrary to the benefits of using inclinometers [24], this monitoring system presents limited precedents in the literature of SHM of bridges [47,48]. Among the reasons given by Huseynov [32] to explain the lack of use of the inclinometers is the lack of sensor technology, low-frequency sampling, and the cost of the current inclinometers.

To solve the aforementioned drawbacks of inclinometers, low-cost sensors can be used. In fact, Micro-Electro-Mechanical Systems (MEMS) accelerometers have revolutionized measuring applications with reduced size and price. It should be noted that other low-cost MEMS accelerometers were already published for SHM applications in the literature, for example:1-Grimmelsman et al. [49] investigated the use of a low-cost accelerometer (ADXL335) with a sampling frequency of 100 HZ. Both the performance and functionality of this accelerometer was compared to those of standard instrument-grade accelerometers (PCB 393A03 and 3741E122G). From this study a difference of 16.2% between the acquired acceleration amplitude of the developed accelerometer and the commercial sensors was obtained.2-Ozdagli et al. [50] developed a low-cost, efficient wireless intelligent sensor (LEWIS) with a sampling frequency of 100 Hz using a low-cost sensor MPU6050. This device was used in a number of laboratory experiments and its results were compared to those of a linear variable differential transformer (LDVT) sensor and a commercial accelerometer (PCB 3711B1110G).3-Meng et al. [51] presented a low-cost acquisition system based on an accelerometer LSM9DS1 and a Raspberry Pi. A laboratory experiment was done to verify this accelerometer, and the results of the developed system were compared to those of a commercial accelerometer (PCB 356B18). The acceleration amplitude of this device was 6.07 percent different from the data of the commercial accelerometer.4-Bedon et al. [52] developed a low-cost self-made accelerometer with a maximum sampling frequency of 256 Hz with post synchronization capability using the MEMS chipset KXR94-2050. The feasibility of the developed accelerometer was verified in several laboratory experiments and the acquired data were compared with those of a commercial accelerometer (PCB 356A16) in a field test.

It is known that the measured acceleration of a uniaxial accelerometer divided by the gravitational force of the earth shows the sine of the tilt angle of the accelerometer. However, titling measurement using this technique does not have a high resolution. In fact, a sudden movement or vibration to the structure will result in a substantial acquired inclination. To solve this problem, MEMS sensors are typically coupled with gyroscopes. MEMS gyroscopes measure the angular rate by Coriolis acceleration, enabling the rotational speed measurement [53]. The main drawback of the gyroscopes is bias instability or Flicker noise [54]. Bias instability is the measurement of the bias drift over time while operating at a constant temperature. This drift is due to the inherent noises of the circuit and the components’ imperfections. The bias instability issue can be fixed through different ways of coupling the calculated inclination of the accelerometer with the gyroscope. In this scenario, the constant drifting of the gyroscope is fixed by the accelerometer’s measurements. Almost all current MEMS inclinometers use sensor fusion capability to improve the individual drawbacks of the accelerometer and the gyroscope. In addition, the negative impacts of a sudden movement of the accelerometer estimations are controlled with the gyroscope measurements [55]. Faulkner et al. [56] presented a SHM application for identifying the quasi-static performance of a bridge under traffic loading using rotational measurements. This work shows that accelerometer and gyroscope output fusion utilizing the Kalman filter can enhance rotational measurements. To validate the application of this methodology, a field test was carried out on an operational single-span railway bridge. The acquired rotation measurements of the proposed device were then compared with a reference vision-based measuring system. This work concluded that acquired rotations from accelerometer and gyroscope sensor fusion had a better correlation with the reference system than those calculated from the accelerometer outputs only.

Allan variance is used to characterize and analyze those noises that drift throughout time in time–domain series [57].

Nowadays, more and more research and methods are being used in the literature to improve the resolution of tilting measurement using MEMS sensors through coupling the estimation of an accelerometer with a gyroscope [33]. Two of the most-used methods are complementary filter and Kalman filter. On the one hand, complementary filter averages calculated angles from an accelerometer and a gyroscope with different weights.

On the other hand, it is mentioned by many scholars that the Kalman filter has a high computational requirement [58,59]. Hence, typically, an inclinometer using the Kalman filter has a lower sampling frequency than an inclinometer that uses the complementary filter [55,60].

It should also be noted that MEMS circuits are based on sensitive sensors that measure a change in the environment correlated with time [2]. In other words, the implemented gyroscope and accelerometer of a MEMS circuit are dynamic sensors. Every dynamic sensor experiences inherent dynamic noises.

Nowadays, for controlling low-cost sensors, microcontrollers are overwhelmingly used. There are various types of microcontrollers available such as Arduino, this being one of the most popular ones on the market. It is based on open-access hardware and software [61].

Some examples of these works include:Yan et al. [62], who developed a low-cost wireless inclinometer with a sampling frequency of 20 Hz and reported resolution of 0.0025°, transmitting its data to acquisition equipment up to 2000 m away. This system is intended to monitor the swing of large-scale structures.Ruzza et al. [60], who introduce a low-cost inclinometer based on the Arduino technology and MEMS circuits with RMS error of between ±0.162 and ±0.304°.Andò et al. [63], who proposed a low-cost multi-sensor system to investigate the structural response of buildings. This system is based on Arduino technology and uses the XBEE interface for wireless communication.Hoang et al. [64], who developed a highly effective robust orientation system for inclinometers in static and dynamic cases. The reported RMS error of static and dynamic tests were 0.106 and 0.091 degrees, respectively.Khan et al. [65], who presented a low-cost inclinometer with a movable electrode. The movable electrode works as a pendulum inside a parallel plate capacitor. The resolution of this inclinometer is reported as 0.38 degrees [65].Woon Ha et al. [66], who proposed a low-cost wireless MEMS inclinometer with a measurement error of 0.04 degrees for an inclination of 0.44 degrees. This inclinometer is meant to estimate the ground movement.

However, the Arduino technology has a few drawbacks: 

(1) Price: Even though these microcontrollers have not been updated or improved lately, their price tag has not decreased by much.

(2) Internet: To connect an Arduino to the Internet or a hotspot, extra parts are needed. 

(3) Basic Model Quality: Low memory size and CPU speed are offered in the basic low-cost versions of Arduino products. 

On the contrary to Arduino systems, NodeMCU is a new microcontroller based on Internet of Thing (IoT) that can be programmed using the Arduino platform and connected to available Wi-Fi hotspots [67]. Its performance can be compared with the Arduino Due. However, it has a fraction of the price of the Arduino Due. The cost of the NodeMCU is €3.95 [68], while the Arduino Due is at least €36.95 [69].

The literature review shows no accurate, low-cost inclinometers based on the Arduino or NodeMCU technology that could be used in SHM of bridges due to the special peculiarities of this type of monitoring [32]. It is indicated by many scholars (such as [24]) that the needed tilt accuracy must be lower than 0.05 degrees. It is shown that the movement of a loaded truck on a bridge induces an inclination with an order of magnitude of 0.2 degrees in the mid-span of a 20 m-long simply supported bridge [24]. In fact, the current low-cost inclinometers share a few drawbacks such as:

(1) Building instructions: these inclinometers are not open hardware and for that reason, the industry cannot use those works to produce a low-cost inclinometer. 

(2) Accuracy: the accuracy of most of them is not comparable with those presented in Table 1. 

(3) Resolution: the resolution of most of the low-cost inclinometers with a high sampling frequency is not acceptable for SHM of bridges. To fill these gaps, this paper presents, for the first time in the literature, a low-cost adaptable reliable angle-meter (LARA) system for SHM of bridges. LARA is a low-cost wireless inclinometer based on an IoT-based microcontroller (NodeMCU) technology with an accuracy of 0.003 degrees based on the performed experiments of this paper.

This paper studies the complementary filter for combining the measurements of the accelerometer and gyroscope of a low-cost MEMS circuit. This way, no heavy computational system is needed for calculations of the Kalman filter. Consequently, the sampling frequency will be higher than when the Kalman filter is used. LARA is based on MEMS technology and uses the complementary filter to couple the outputs of its accelerometer and gyroscope calculated angles.

In addition, to build an inclinometer with higher accuracy, better resolution and lower noise density, this paper develops a custom-designed Printed Circuit Board (PCB) containing five low-cost aligned MEMS MPU9250 chipsets, each of one incorporating a gyroscope, an accelerometer and a magnetometer. It should be noted that MEMS accelerometers are frequently influenced by inherent noise accelerometers [70]. Intrinsic noise is created by components within a circuit (such as resistors and semiconductors) [71].

By averaging the outputs of several dynamic sensors, the primary under-study captured signal is not altered. However, the inherent dynamic noises (also known as intrinsic noise) of the sensors will be averaged by the number of combined sensors. As the magnitude of these dynamic noises decreases, smaller dynamic changes like acceleration or angular speed can be detected. To illustrate the beneficial effect of combining several gyroscopes and accelerometers, the Allan variance of different sensor combinations is compared with the results of a single MPU9250 working an inclinometer. In addition, laboratory experiments are carried out and the results are compared with those reported by a commercial inclinometer (HI-INC). The obtained results illustrate that using sensor combination can turn a low-cost sensor resolution comparable with a high-quality commercial solution. Finally, LARA is verified by being allocated on a simply supported beam, and its estimations are backed up by hand calculations.

This paper is organized into five sections. In the second section, first, the commercial inclinometer used as a reference value in the paper (HI-INC) is introduced. Then, the proposed low-cost solution (LARA) is presented. In the third section, resolution experiments analyzing the beneficial effect of a similar sensor combination are illustrated. The fourth section is dedicated to laboratory experiments verifying the accuracy and resolution of the LARA, plus the results and discussions. Finally, in the last section, the main conclusions of the work are drawn.

## 2. Control System and the Proposed Inclinometer

In this section, first, the main characteristics of a control system for measuring inclination are drawn. Then, the proposed inclinometer of this work is presented. In addition, the needed equipment and the setting up protocol of the control system and the proposed inclinometer are reviewed.

### 2.1. Control System Description

In this study, BeanDevice^®^ Wilo HI-INC (Figure 1a), an ultra-low-power (ULP) biaxial WIFI inclinometer, was used as the high-accuracy controlling system. This device contains a built-in data logger that can store up to 5 million data logs with a maximum wireless range of 200 m. Regarding angle measurements, it combines a high-performance inclinometer sensor and a 24-bit delta–sigma analog-to-digital converter, making it possible to have a high-level accuracy of ±0.003° for ±15° and a resolution of 0.001°. In addition, the body of the HI-INC inclinometer is composed of a lightweight aluminum casing with waterproof capability [72]. The program used for data acquisition is a commercial solution promoted by the BeanDevice company and costs €350. Taking the price of this inclinometer from Table 1 and the needed commercial software for data acquisition into account, the whole solution costs around €1000.

The data acquisition program of the BeanDevice company acquires and in real-time illustrates the *X* and *Y* axis’ inclinations (Figure 1b). Finally, it should be noted that the settings of data acquisition (such as sampling frequency) can be modified from the main menu of the commercial program (Figure 1c).

It should also be noted that the inclinometer HI-INC has been recently used as a wearable gadget for frequency analysis of a walking pedestrian for the assessment of lightweight glass slabs (see e.g., [73]).

### 2.2. Low-Cost Adaptable Reliable Angle-Meter (LARA) System

In this section, the hardware architecture of the proposed inclinometer is presented. Then, the software part of this system is explained and shown.

#### 2.2.1. Hardware Architecture of LARA

This paper proposes multiple combinations of gyroscopes and accelerometers for producing a more accurate inclinometer. To this end, five chipsets of MPU9250 are engineered together on a single PCB and synchronized using a multiplexor (TCA9548A). To avoid the problems of a manual fabrication (such as nonalignment of the circuits, time-consuming process of aligning, soldering, and sensor quality control and size), the PCB of LARA was designed and produced to satisfy the delicacy of current project measurements. In addition, the required components of LARA are soldered to the PCB using machine assembly. Figure 2a,b shows the produced sensor and its blueprint. It should be noted that LARA can be assembled by hand using available commercial MPU9250 circuits and a TCA9548A multiplexor. Figure 2c shows the Fritzing [74] sketch of the system. The cost of a LARA made by connecting five MPU9250 and TCA9548A and a bulk company-produced PCB with assembled components is around €37 and €51, respectively.

As shown in Figure 2a,b, LARA has four output ports. These wires should be connected to a microcontroller to power up the sensors, acquire the sampled data, and convert the gyroscope and the accelerometer to tilt and pitch inclination. The used microcontroller of this paper is NodeMCU and shown in Figure 2d. This low-cost open-source Internet of Things (IoT) platform runs on the ESP8266 chipset. ESP8266 is a low-cost Wi-Fi microchip with Internet protocol suite (also known as TCP/IP) capability [75].

#### 2.2.2. Software Architecture of LARA

In this section, the software used for this project is presented.

Arduino platform: NodeMCU is first programmed using the Arduino platform. This program first estimates the angle in real-time from each of the individual MPU9250 chipsets. Then, the formulas for calculating the rotation using a triaxial accelerometer for the *X* and *Y* axes are presented in Equation (1) and Equation (2), respectively.
(1)$$angleaccX=\tan^{-1}\left(\frac{accY}{\sqrt{accZ^{2}+accX^{2}}}\right)\times \left(\frac{360}{2\pi }\right)\;$$
(2)$$angleaccY=\tan^{-1}\left(\frac{accX}{\sqrt{accZ^{2}+accY^{2}}}\right)\times \left(\frac{360}{2\pi }\right)$$

In Equation (1) and Equation (2), angleaccX and angleaccY are the calculated angles from the acquired data of a MPU9250 accelerometer around the *X*-axis and *Y*-axis, respectively. The accX, accY and accZ represent the obtained acceleration data of the *X*, *Y* and *Z* axes. Then, using a complementary filter, the calculated angle from the accelerometers and the acquired data of the gyroscopes are combined. Equation (3) and Equation (4) present the used complementary equation for the fusion of the gyroscope and the accelerometer results for measuring the rotation around *X* and *Y* axes, respectively.
(3)$$angleX=\left(0.96\times \left(angleX_{0}+gyroX\times time\right)\right)+0.04\times angleaccX$$
(4)$$angleY=\left(0.96\times \left(angleY_{0}+gyroY\times time\right)\right)+0.04\times angleaccY$$

In Equations (3) and (4), angleX and angleY are the final calculated rotations around the *X* and *Y* axes, respectively. The $angleX_{0}$ and $angleY_{0}$ are the estimated angle of the system from the previous measurement. In the initiation of the data acquisition, it should be noted that this value equals zero. After that, it represents the rotation progress. GyroX and GyroY represent the measured angular speed of the gyroscope for the *X* and *Y* axes, respectively. The *time* presents the interval time between two measurements. Further analysis of these equations shows that the angle calculated from the accelerometer is multiplied by a smaller coefficient than that of the gyroscope [76]. This low coefficient factor of angleacc is for mitigating the impact of environmental vibrations (also known as cross-talk of vibration) and can vary between 0.02 and 0.05 [77].

These equations are repeated for every MPU9250 chipsets of LARA. Then, the inclination values of the five chipsets are averaged separately for the *X* and *Y* axes. It should be noted that this code makes the implemented accelerometers and gyroscopes of LARA sample data and estimate the angles in a synchronized way. Finally, using the already introduced service set identifier (SSID) and the router’s password in the Arduino code and the averaged results of the *X* and *Y* axes are transmitted to a made-up server client by the built-in ESP8266 chipset. LARA prints a server address and a port number at this stage on the serial port of the Arduino. This information should be noted, and LARA can be detached from the programming computer. After this, the sensor can be disconnected from the PC and plugged into any available USB power break.

Virtual serial port: After connecting LARA to a USB power source, the data sampling function initiates automatically. This chipset’s TCP/IP capability helps this sensor provide its outputs on a local server. A computer connected to the same SSID as LARA can stream the sampled data by introducing the noted server address and port number of LARA. In order to acquire the sampled data and have a real-time graphical representation of the LARA inclination, a virtual serial port application is used [HW [78]]. This free software needs the server address and the port number of LARA and creates a virtual serial port communication connection between LARA and a windows-based computer. By selecting the provided virtual port of the HW software on the Arduino platform, LARA’s sampled data can be streamed or graphed just when the sensor is connected to the computer. A computer can indeed be connected physically to several sensors, but with the HW virtual serial port, up to 99 devices can be wirelessly attached to a single computer.

Data acquisition: Unlike the Arduino platform, free commercial software like SerialPlot [79] can represent the sampled data in real-time in a graphical interface and save the data with the date and timestamp of data acquisition. The presented flowchart in Figure 3 shows the steps of real-time inclination acquisition using LARA.

## 3. Statistical Representation of Combining Dynamic-Sensor Theory

This section first studies the effect of sensor combination on the noise density, standard deviation and resolution of angle measurements. Then, the Allan variance and its importance in evaluating the noise density of inclinometers in the literature are explained. Finally, the Allan variances of several combined sensors are presented.

### 3.1. Noise Reduction of Inclinometers

This section explains an experiment that leads to combining up to five similar circuits (MPU9250) for reducing overall dynamic (harmonic) noises. During this experiment, the inclinometers were placed in a quiet environment, far away from crowds and with reduced induced ambient vibrations. The aim of this experiments is to measure and evaluate the pure noise ratio of different combined inclinometers.

It was noticed that the average value of outputs of several aligned synchronized inclinometers has lower noise density than the those of a single one. The standard deviation of up to five combined inclinometers is presented in Figure 4a.

The analysis of Figure 4a shows that the higher the number of sensors considered, the lower the noise density of their averaged measurements that the more combined inclinometers have a lower noise density. The reason behind the beneficial behavior of combined inclinometers is within the inherent dynamic noises of the produced accelerometers and gyroscopes chipsets. Figure 4b shows the frequency domain illustration of the performed experiment. Data transformation from the time domain to the frequency domain is done using fast Fourier transformation (FFT). The analysis of Figure 4b shows that the magnitude of the dynamic noises of the averaged values of a set of sensors made from combined inclinometers is lower than that of a single one. It can be seen that on 1 Hz the measured noises for a single inclinometer and five combined inclinometers are 3.9 × 10^−4^ and 2.6 × 10^−4^ degrees, respectively.

These results led to investigating the beneficial impact of dynamic sensor combinations. Analyzing the individual outputs, the five used MPU9250 sensors showed that every single sensor has unique dynamic noises.

Furthermore, a single output that includes the averaged inherent noises of all individual inclinometers plus the understudy signal (the rested situation or sets of dynamic movements) is obtained by averaging the outputs of several inclinometers. Since the understudy signals are not dependent on the characteristics of the inclinometers, they have not impacted the FFT process. The FFT highlights the most repeated signals (the understudy ones) and undervalues those that are repeated less, such as the inherent individual noises of the sensors. By improving the noise density, the inclinations that in the first place were smaller than the noise density of the sensor can now be detected due to the improved noise level.

### 3.2. Study of Allan Variance

Allan variance is typically used to characterize and analyze those noises that drift throughout time in the time domain series [57]. In fact, Allan variance quantifies the measurement variance of a sensor across different timescales. Contrary to frequency-domain noise evaluation methods such as spectral noise density (ND), Allan variance is a time-domain evaluating tool of different noise sources (such as quantization, angle random-walk, bias instability, rate random-walk, and rate ramp) [80]. Allan variance shows the progress of a noisy sensor signal over time which can be very useful to identifying the progressive random-walk of a gyroscope instead of ND that quantifies the noise density of an accelerometer [57]. Allan deviation is more commonly used as the square root of Allan variance [81]. The available acquired inclination acquisition data for measuring the standard deviation of the previous subsection was used for the Allan variance and deviation calculations. Figure 5a,b shows the log–log plot of Allan variance and Allan deviation of a single and up to five synchronized inclinometers, respectively.

Analysis of Figure 5a shows that the higher the number of combined inclinometers, the lower progressive the noise is. For example, the first calculated value of the Allan variance (Figure 5a) of a single inclinometer and five combined ones are 1.45 × 10^−2^ and 6.7 × 10^−3^, respectively. The beneficial effect of additional synchronized sensors can also be seen in the Allan variation presented in Figure 5b. It is indicated in the literature [82] that various noise types (such as white noise, flicker noise and random) can be detected from the log Allan deviation plot. The detection of different noise types from a gyroscope output is presented in [83].

This section showed that sensor combination decreases noise magnitude in both the time domain and the frequency domain.

## 4. Laboratory Experiments

In this section, LARA’s measurement accuracy is evaluated by comparing its results with the estimations of a HI-INC inclinometer in four tests. Then, the combinatory analysis presents the accuracy improvement of inclination measurement of up to five combined inclinometers. Finally, the accuracy and resolution of HI-INC and LARA are validated by performing four load tests on a simply supported aluminum beam.

### 4.1. Accuracy Evaluation

In this section, the experimental tests targeted at verifying the accuracy of LARA are shown. In order to make sure that LARA and the commercial inclinometers measure the same inclination, LARA was glued on top of the HI-INC clinometer. Then, the HI-INC was connected to the rigid metallic plate using its magnetic plate. After that, the metallic plate was connected to a rotational device (Figure 6). By rolling the small gear of this rotational device, the connected stiff plate rotates. Then, the induced rotations were measured by the LARA and HI-INC. Finally, the outputs of LARA for the tests were compared with those estimated by the HI-INC inclinometer.

Table 2 presents the results of the carried out experimental tests. This table takes in the following information collected in columns. (1) N: four tests are carried out for evaluating the accuracy of LARA in different inclinations; (2) HI-INC: the estimated inclination by HI-INC inclinometer; (3) LARA: the measured inclination of LARA; and (4) difference: the absolute difference of LARA and HI-INC measurements.

The analysis of Table 2 shows that the difference in LARA measurement from the HI-INC is related to the induced inclination. In fact, it was seen that for more than five degrees of change, the difference of LARA from HI-INC was higher than 0.1 degrees. For that reason, their data are not included in Table 2. Therefore, it can be concluded that the accurate measuring range of LARA is up to four degrees. This range is accurate enough for the target application of bridge monitoring, as in these kinds of structures increments of rotation higher than 0.5 degrees are not expected [24,84].

### 4.2. Combinatory Analysis

In order to study the difference of the measured values from the reference sensor for different inclinometer combinations, a combinatory analysis was performed. This evaluation illustrates the maximum and minimum envelope difference from the commercial inclinometer for all the possible sensor selections from the five available inclinometers. The maximum and minimum values of the increasing number of inclinometers are shown in Figure 7, Figure 7a (one sensor), Figure 7b (two sensors), Figure 7c (three sensors), and Figure 7d (four sensors). It is to be noted that in all of these figures, LARA shows the estimation calculated by combining the results of five inclinometers together.

The analysis of Figure 7 shows that the accuracy of the whole system is directly influenced by the number of combined inclinometers. For example, the minimum accuracy of a single inclinometer (the max difference from HI-INC) is 0.0557 degrees for an induced inclination of 0.9996. However, for the same experiment, LARA showed a measurement difference of 0.0381 from HI-INC. As expected, the higher the number of sensors, the better the accuracy of the modular system. It is essential to note that the minimum difference from HI-INC estimations reported in Figure 7a does not correspond to the measurement of a single sensor for all four experiments. It can also be seen that the distance between the minimum and the maximum differential from HI-INC values decreases with a higher number of combined sensors. In fact, having a lower range of possible errors can help making the final product more reliable. This reliability is very important when an inclinometer has a high sampling frequency. This way, optimizing filters (such as different Kalman filter formulations [85]), which could alter the primary signal and slow down the acquisition speed, are no longer necessary.

### 4.3. LARA Resolution and Accuracy Verification Using a Beam Model

In order to present the resolution and accuracy of LARA more clearly, a load test was performed on a small-scale beam with a length of 1.24 m. This section compares the slope estimation of two sensors (LARA and H-INC) located on the support of a simply supported aluminum beam model under a point load of 467 gr (4.58 kN) with hand calculation of slope at the beam edges. It should be mentioned that, for this test, LARA was again mounted on the top of the HI-INC.

This test was carried out using a U-shaped aluminum profile with section dimensions of 25 × 25 × 3 × 3 mm. The effective length of the beam model, which is the distance between the null axis of its support, is fixed as 1080 mm.

The test aim was to read the maximum slope of the beam model deck under a known applied load on the mid-span. The maximum slope at the supports can be calculated by Equation (5). Therefore, LARA and HI-INC were attached to achieve this objective on top of the beam model support. First, LARA and HI-INC worked for a while without any loads (Figure 8a) and their estimations were acquired. Next, the point load was set on the mid-span of the beam model (Figure 8b,c), and then another data acquisition process was carried out to measure the slope of the beam by LARA and HI-INC. It is essential to mention that this test was repeated three times.

The used formula for calculating the slope of a simply supported beam with a load located on its midspan by hand is presented in Equation (5) [86].
(5)$$\Delta \theta_{1}=\frac{P\times L^{2}}{16\times E\times I}$$

In Equation (5), ∆*θ*_1_ (radians) is the maximum slope at the supports, *P* is the value of the applied load at the mid-span, *L* is the effective beam length, *E* (69,637.05 MPa) is the beam elasticity module, and *I* (12,853.08 mm^4^) is the beam moment of inertia. ∆*θ* is then calculated as 0.000373 radians. This value corresponds to 0.021372 degrees on inclination. The comparison of the estimated values of LARA and HI-INC with those of the hand calculations is presented in Table 3. It should be noted that this test was repeated three times (Table 3) to check the accuracy of the developed inclinometer.

The analysis of Table 3 shows that the accuracy of LARA based on these experiments is less than 0.002 degrees. Further study of Table 3 illustrates that the accuracy of HI-INC is around 0.005 degrees. In fact, this is very close to the accuracy value detailed in its datasheet (±0.003° for ±15° version) [87]. This value validates the accountability of the performed experiment. Therefore, having accuracy in the range of 0.05 degrees makes LARA applicable for the SHM of bridges.

Another experimental test was carried out on this beam model (Figure 8a) using a heavier weight (21.942 N). In this experiment, instead of putting the weight only on the midspan, the weight was set on various beam locations. Then, the support slope was measured using HI-INC and LARA. Finally, the sensors’ measurements are compared with the hand calculations [86]. Figure 9 presents the slope measurement comparison of HI-INC and LARA with the hand calculation values. It is vital to mention that this experiment is carried out on the same beam model presented in Figure 8a. As shown in this figure, the inclinometer is mounted on a pinned support.

Analysis of Figure 9 shows that LARA has a maximum measured difference of 0.003 degrees from the hand-calculated slope. In addition, it can be seen that LARA has a closer trend to the hand-calculated values compared to those of HI-INC.

It should be noted that LARA can be used in static load tests aiming to identify the location of structural damages which had altered the influence line of a bridge [24].

It is interesting to compare LARA’s final price (€54.95) with those presented in Table 1. It should be noted that comparing an academically developed device with a commercial alternative is not fair. However, the most critical contribution of current work is developing a low-cost, accurate device, and for that, this comparison is needed. It can be seen from Table 1 that HI-INC, ZCT-CX09 and DNS have a resolution of 0.003 degrees. Therefore, LARA can be compared with them. Figure 10 presents the price comparison of these inclinometers.

Analysis of Figure 10 shows a significant difference between the price of LARA and inclinometers with the same resolution. LARA is 12, 6 and 6 times cheaper than HI-INC, ZTC-CX09 and DNS inclinometers, respectively. Also, it does not need extra paid commercial software for data acquisition, and it is based on open-source software and hardware.

## 5. Conclusions

Lately, the implantation of inclinometers for the SHM of bridges is receiving a lot of attention from engineers and researchers. In fact, unlike accelerometers, inclinometers can enable easy evaluation of both the location and severity of structural damage. This characteristic makes them suitable for the long-term structural health monitoring (SHM) of bridges. In addition, the deflection of a structural member can be easily estimated by using inclinometers. However, current inclinometers’ high prices have limited their use. There is gap in the literature for the development of a low-cost inclinometer for the long-term SHM of bridges with a low budget for their health assessments.

To fill these gaps, in this paper, a low-cost adaptable reliable angle-meter (LARA) system is presented. LARA is a low-cost wireless IoT-based inclinometer with a sampling frequency of 250 Hz. It consists of five inclinometers (MPU9250), a multiplexor and an IoT-based microcontroller (NOSEMCU). Every inclinometer combines the acquired values of its accelerometer and its gyroscope using a complementary filter. The main novelty of LARA is combining the results of five aligned inclinometers for reducing the inherent noise density of individual accelerometers and gyroscopes of LARA.

In order to validate the assumption of noise reduction and signal improvement for inclination measurements using the averaged results of several aligned inclinometers, four laboratory experiments were carried out. The results of these tests show that averaging the values of a number of aligned accelerometers reduces the noise density of the frequency domain representation of a vibration acquisition experiment. In addition, it is shown that the Allan variance and deviation of a system consisting of five aligned inclinometers are significantly better than those of a single inclinometer.

To validate the accuracy of LARA, an experimental test was designed to validate the measurement of LARA in a rotation range of zero to four degrees. In this test, the acquired data of LARA was compared with the values of a commercial inclinometer. It is seen that LARA presents up to 0.04 and 0.07 degrees of difference in tests with one and four degrees of inclination, respectively. In the SHM of bridges, the structure rarely expects to experience a slope of more than 0.5 degrees.

In addition, in order to compare the accuracy of the used commercial inclinometer and LARA, a load test was performed on a beam. In this test, the reported values of the commercial inclinometer were compared with LARA’s. It was shown that LARA estimated the theoretical slope with less than 0.003 degrees of difference from the hand-calculated values. However, HI-INC showed an accuracy higher than its datasheet data with a magnitude of ±0.005°.

This high accuracy of rotation estimation makes LARA applicable to the SHM of bridges and damage detection techniques tailored for inclinometers. However, in future research, LARA needs to be validated in a load test application of a bridge.

## Figures and Tables

**Figure 1 sensors-22-05605-f001:**
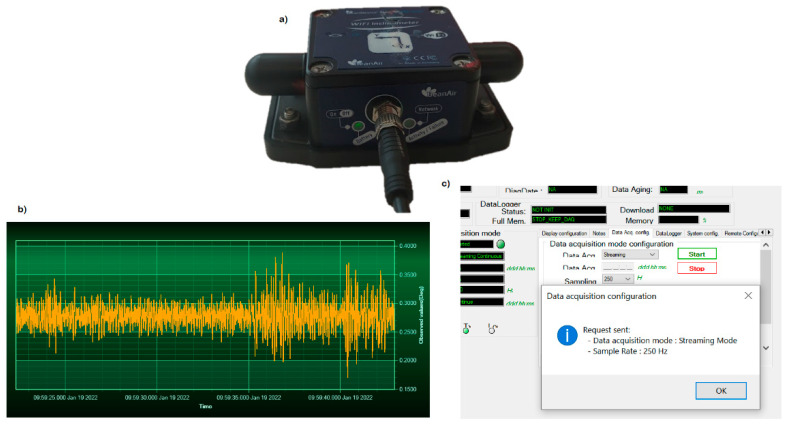
(**a**) HI-INC biaxial inclinometer, (**b**) inclination streaming over *X* axis and (**c**) sampling frequency rate.

**Figure 2 sensors-22-05605-f002:**
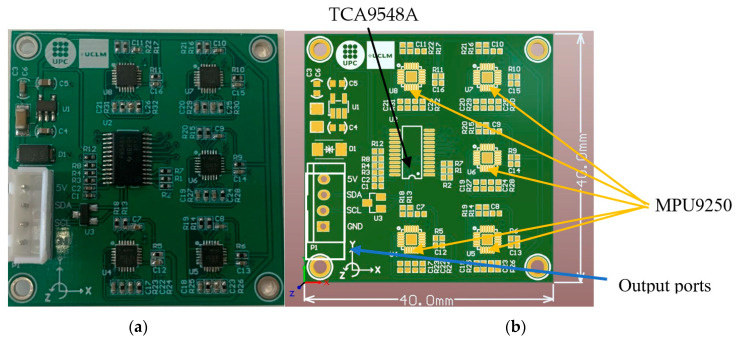
Illustration of LARA: (**a**) the produced product, (**b**) the blueprint of the designed PCB, (**c**) the Fritzing sketch of the system, and (**d**) the NODE MCU microcontroller.

**Figure 3 sensors-22-05605-f003:**
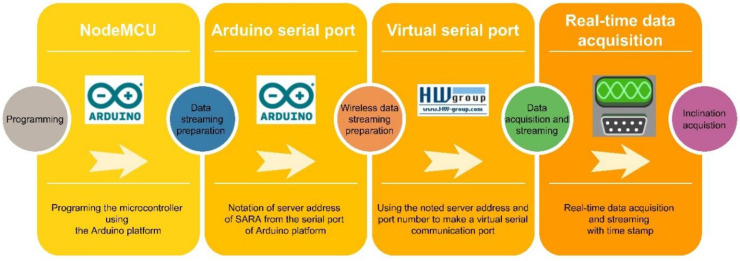
The required steps of real-time wireless inclination acquisition using the LARA inclinometer.

**Figure 4 sensors-22-05605-f004:**
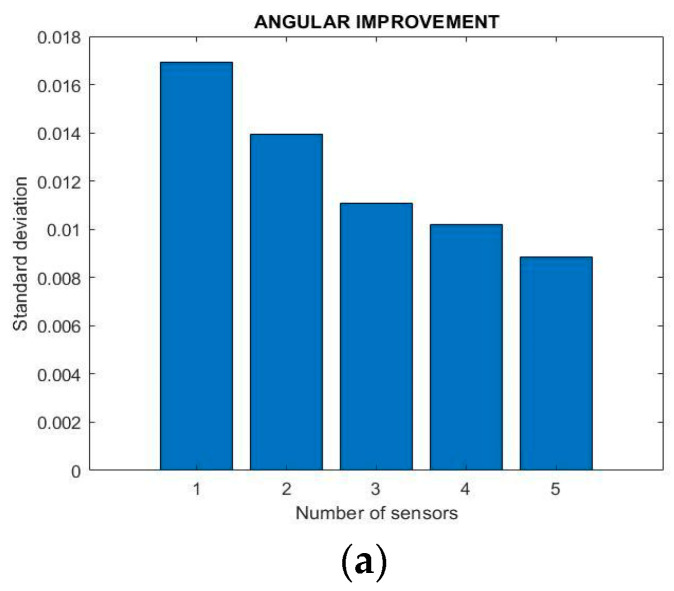
Representation of the noise ratio of a single and up to five combined inclinometers using (**a**) standard deviation and (**b**) noise density in the frequency-domain.

**Figure 5 sensors-22-05605-f005:**
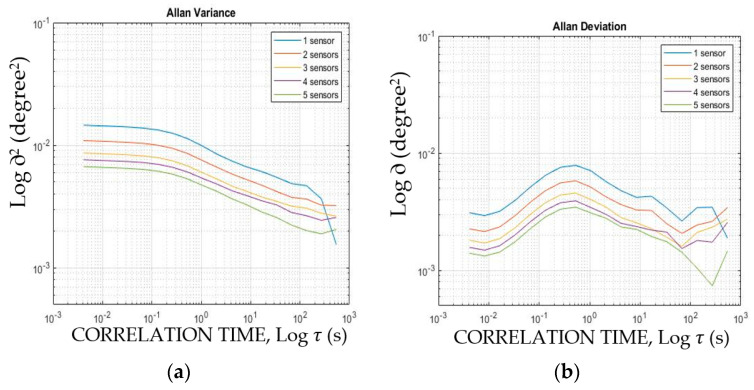
Quantifying the noise progress of various inclinometer combinations in the time domain using (**a**) Allan variance and (**b**) Allan deviation.

**Figure 6 sensors-22-05605-f006:**
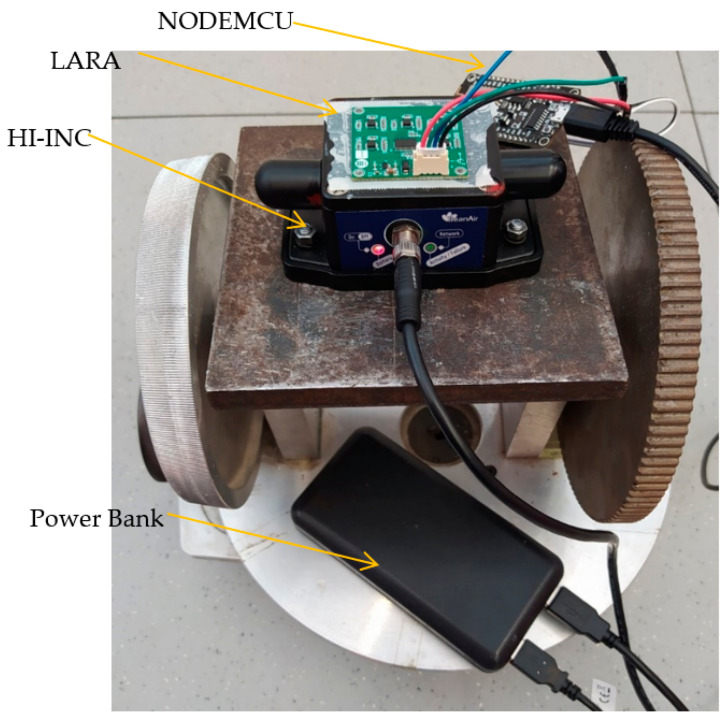
Test setup intended for comparing inclination estimation of LARA with HI-INC.

**Figure 7 sensors-22-05605-f007:**
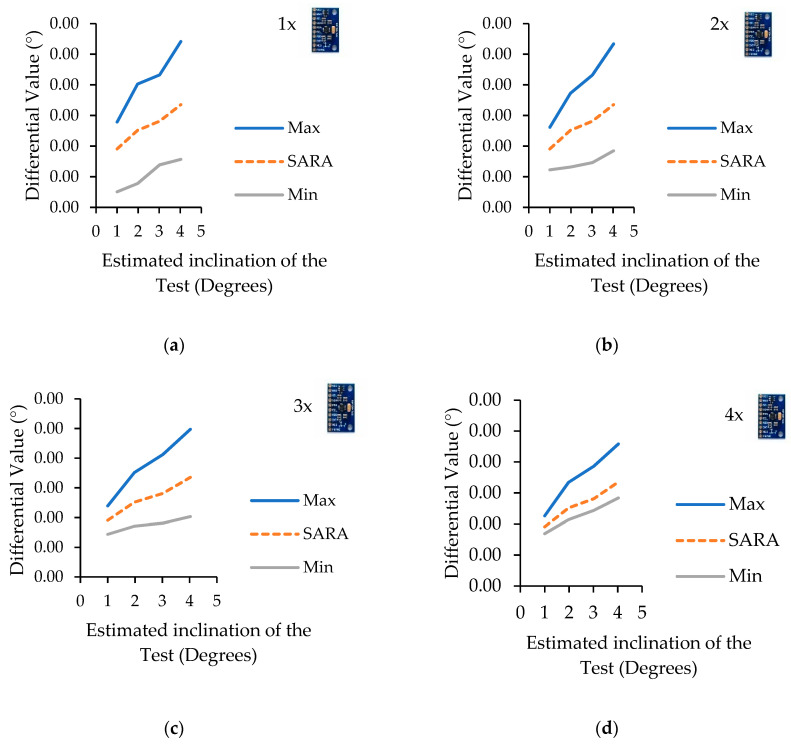
Estimated measured inclination difference for a different number of combined inclinometers from HI-INC estimations. One sensor (**a**), two sensors (**b**), three sensors (**c**), and four sensors (**d**).

**Figure 8 sensors-22-05605-f008:**
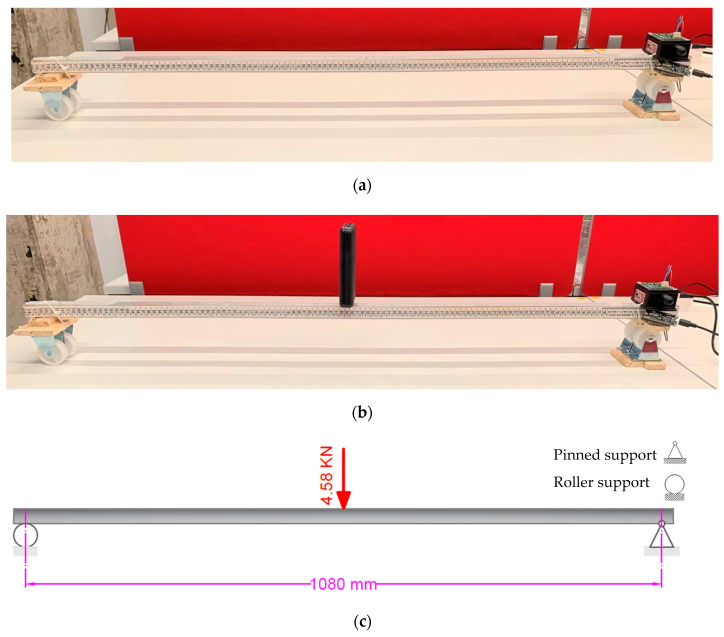
Load test of a beam model: (**a**) test setup, (**b**) load test, and (**c**) sketch of the load test.

**Figure 9 sensors-22-05605-f009:**
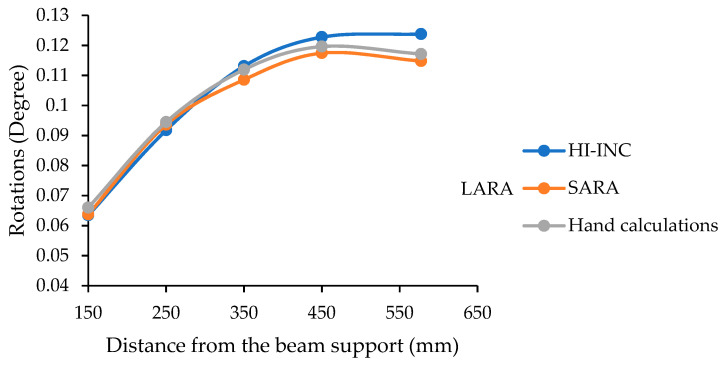
Support slope of a simply supported beam under a point load located on various spots.

**Figure 10 sensors-22-05605-f010:**
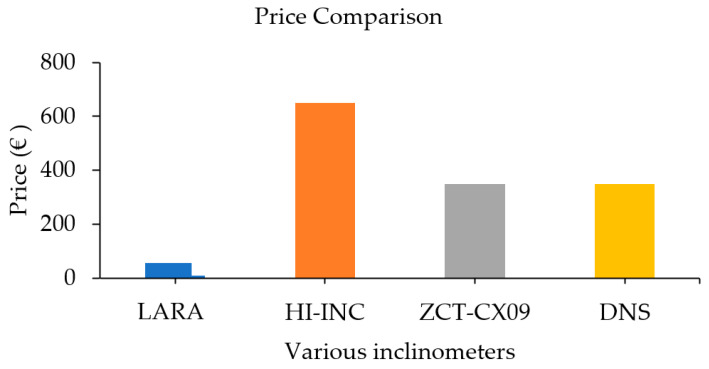
Price comparison of LARA with traditional commercial inclinometers with a resolution of 0.003 degrees.

**Table 1 sensors-22-05605-t001:** Characteristics of some of the commercially available inclinometers.

Model	Measurement Range (Degrees)	Resolution (Degrees)	Sampling Rate (Hz)	Price (€)
ZEROTR-ONIC	±0.5°	100 × 10^−5^°	10	3950
JDI 200	±1.0°	10 × 10^−5^°	125	2250
T935	±1.0°	6 × 10^−5^°	10	1696
ACA2200	±0.5°	10 × 10^−5^°	20	710
HI-INC	±15.0°	100 × 10^−5^°	100	650
ZCT-CX09	±15.0°	100 × 10^−5^°	8	350
DNS	±85.0°	300 × 10^−5^°	100	348

**Table 2 sensors-22-05605-t002:** Accuracy comparison of LARA with HI-INC.

N	HI-INC (Degrees)	LARA (Degrees)	Difference (Degrees)
1	0.9996	0.9615	0.0382
2	1.9770	1.9267	0.0503
3	3.0180	2.9618	0.0563
4	4.0254	3.9583	0.0671

**Table 3 sensors-22-05605-t003:** Comparing the inclination estimation of LARA and HI-INC.

Number of the Experiments	Hand Calculation Slope (Degrees)	LARA Difference (Degrees)	LARA (Degrees)	HI-INC Difference (Degrees)	HI-INC (Degrees)
1	0.021372	0.001613	0.022985	0.002447	0.018925
2	0.021372	0.002316	0.023688	0.000853	0.020519
3	0.021372	0.001362	0.022734	0.005196	0.016176

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
