# Peer review of "A Novel Wireless Low-Cost Inclinometer Made from Combining the Measurements of Multiple MEMS Gyroscopes and Accelerometers"

_sensors, 2022, doi:10.3390/s22155605_

Round 1

Reviewer 1 Report

The manuscript topic is of interest and falls within the scope of Journal.

Few improvements are recommended.

1) there are several low cost solutions in literature for structural health monitoring, see for example JSAN | Free Full-Text | Prototyping and Validation of MEMS Accelerometers for Structural Health Monitoring—The Case Study of the Pietratagliata Cable-Stayed Bridge | HTML (mdpi.com)

and many others. It would be good and realistic to spend few words.

2) the presented sensor is similar to that applied in JSAN | Free Full-Text | Uncoupled Wi-Fi Body CoM Acceleration for the Analysis of Lightweight Glass Slabs under Random Walks (mdpi.com)

for another research application. Please comment in the revised document

3) please define the number of test repetitions 

4) figure 8 (c) should be better described. A legend would be helpful. Units of measure should be described. There is no control on model and simulation.

5) please better clarify the test support for experiments summarized in figure 9

Author Response

The authors sincerely appreciate the positive comments of the two Reviewers and are very grateful for their suggestions and observations. These comments certainly have improved the quality of the paper. In addition, they gave us valuable hints on our future research. Detailed responses to each of the reviewer's comments are provided in the following lines. The answers of the authors are highlighted in green, the new information is in blue and the excluded information in red.

Reviewer 2 Report

General:

This is a good submission about an interesting subject with good chances for direct, field applications. The introduction is excellent. After a few clarifications, it can be published in Sensors.

Please include in the introduction also an analysis of this new paper, very much pertinent to this subject which was published by  Faulkner et al.

https://doi.org/10.1007/s13349-020-00400-9

in  Journal of Civil Structural Health Monitoring in 2020. 

Detailed problems:

add among keywords "structural health monitoring"

page 2: 

...Structural System Identification (SSI) ...

Avoid using shortcut SSI as it a well known short-cut for Soil-Structure Interacion. Just write in full Structural System Identification

Ref. 33 - add name of the journal/book (?) etc.

English language issues:

The last sentence of the abstract:

LARA's affordability and high accuracy make it applicable for structural damage 27 detection and locating in bridges using inclinometers.

should be: makes it.     Also, the syntax of this sentence seems to be awkward.

42-43 (...) model of the real structures (digital-twin), which imitate the performance (...).      should be: model (...) which imitates ...

line 61 "clinometers"?

lines 226-241 unwanted material accidentally appeared 

It should be removed  

Author Response

(The authors gave the same response as above.)

Round 2

Reviewer 1 Report

The original document has been largely revised and is suitable for possible publication